# Maternal Infection and Adverse Pregnancy Outcomes among Pregnant Travellers: Results of the International Zika Virus in Pregnancy Registry

**DOI:** 10.3390/v13020341

**Published:** 2021-02-22

**Authors:** Manon Vouga, Léo Pomar, Antoni Soriano-Arandes, Carlota Rodó, Anna Goncé, Eduard Gratacos, Audrey Merriam, Isabelle Eperon, Begoña Martinez De Tejada, Béatrice Eggel, Sophie Masmejan, Laurence Rochat, Blaise Genton, Tim Van Mieghem, Véronique Lambert, Denis Malvy, Patrick Gérardin, David Baud, Alice Panchaud

**Affiliations:** 1Materno-Fetal and Obstetrics Research Unit, Department “Woman-Mother-Child”, Lausanne University Hospital, University of Lausanne, 1011 Lausanne, Switzerland; manon.vouga@chuv.ch (M.V.); leo.pomar@chuv.ch (L.P.); Sophie.Masmejan@chuv.ch (S.M.); 2Department of Obstetrics and Gynecology, Centre Hospitalier-Franck Joly, 97393 Saint-Laurent du Maroni, French Guiana; v.lambert@ch-ouestguyane.fr; 3Paediatric Infectious Diseases and Immunodeficiencies Unit, Hospital Universitari Vall d’Hebron, 08035 Barcelona, Spain; tsorianoarandes@gmail.com; 4Maternal-Fetal Medicine Unit, Department of Obstetrics, Hospital Universitari Vall d’Hebron, 08035 Barcelona, Spain; carlotarodo@gmail.com; 5Institut Clínic de Ginecología, Obstetricia i Neonatologia and BCNatal (Barcelona Center for Maternal-Fetal and Neonatal Medicine), Hospital Clínic and Hospital Sant Joan de Deu, Institut d’Investigacions Biomèdiques August Pi i Sunyer, Universitat de Barcelona, Center for Biomedical Research on Rare Diseases (CIBERER), 08028 Barcelona, Spain; agonce@clinic.cat (A.G.); egratacos@ub.edu (E.G.); 6Department of Obstetrics and Gynecology, New York Presbyterian Hospital Columbia University, New York, NY 10032, USA; aamerr02@gmail.com; 7Department of Obstetrics and Gynecology, University Hospitals of Geneva, 1205 Geneva, Switzerland; isabelle.eperon@gmail.com (I.E.); begona.martinezdetejada@hcuge.ch (B.M.D.T.); 8Faculty of Medicine, University of Geneva, 1205 Geneva, Switzerland; 9Obstetrics and Gynecology Department, Centre Hospitalier du Centre Valais (CHCVs), 1950 Sion, Switzerland; Beatrice.Eggel-Hort@hopitalvs.ch; 10Center for Primary Care and Public Health, University of Lausanne, 1011 Lausanne, Switzerland; Laurence.rochat@huv.ch (L.R.); blaise.genton@unisante.ch (B.G.); 11Maternal-Fetal Medicine Unit, Department of Obstetrics and Gynecology, Mount Sinai Hospital and University of Toronto, Toronto, ON M5G 1X5, Canada; Tim.vanmieghem@sinaihealth.ca; 12Department for Infectious and Tropical Diseases, CHU Hôpitaux de Bordeaux and Inserm 1219, University of Bordeaux, 33000 Bordeaux, France; denis.malvy@chu-bordeaux.fr; 13INSERM CIC1410 Clinical Epidemiology, Centre Hospitalier Universitaire de la Réunion, 97410 Saint Pierre La Réunion, France; patrick.gerardin@chu-reunion.fr; 14School of Pharmaceutical Sciences, Geneva University and Service of Pharmacy, Lausanne University Hospital, 1011 Lausanne, Switzerland; alice.panchaud@chuv.ch; 15Service of Pharmacy, Lausanne University Hospital, University of Lausanne, 1011 Lausanne, Switzerland; 16Institute of Primary Health Care (BIHAM), University of Bern, 3012 Bern, Switzerland

**Keywords:** Zika, congenital Zika syndrome, pregnancy, travelers

## Abstract

In this multicentre cohort study, we evaluated the risks of maternal ZIKV infections and adverse pregnancy outcomes among exposed travellers compared to women living in areas with ZIKV circulation (residents). The risk of maternal infection was lower among travellers compared to residents: 25.0% (*n* = 36/144) versus 42.9% (*n* = 309/721); aRR 0.6; 95% CI 0.5–0.8. Risk factors associated with maternal infection among travellers were travelling during the epidemic period (i.e., June 2015 to December 2016) (aOR 29.4; 95% CI 3.7–228.1), travelling to the Caribbean Islands (aOR 3.2; 95% CI 1.2–8.7) and stay duration >2 weeks (aOR 8.7; 95% CI 1.1–71.5). Adverse pregnancy outcomes were observed in 8.3% (*n* = 3/36) of infected travellers and 12.7% (*n* = 39/309) of infected residents. Overall, the risk of maternal infections is lower among travellers compared to residents and related to the presence of ongoing outbreaks and stay duration, with stays <2 weeks associated with minimal risk in the absence of ongoing outbreaks.

## 1. Introduction

Zika virus (ZIKV) has emerged as an arthropod -borne infection associated with adverse pregnancy outcomes [1]. The risks associated with intrauterine ZIKV infection have been well documented among women living in areas with active ZIKV circulation where the overall risk of severe adverse pregnancy outcomes for exposed foetuses was estimated to range between 5 to 13% [2,3,4]. However, the risks for pregnant travellers with brief exposures remain poorly described. Though transmission has now declined all over the world, epidemic clusters are still being reported [5] with the possibility of emergence/re-emergence in all areas where competent vectors are found [1]. Given the known sexual transmission and the risk for maternal infection at an early stage of pregnancy, several international agencies [6,7] continue to recommend a 2 to 3-month delay prior to attempting conception after returning from areas with ongoing or past ZIKV circulation. As these regions encompass most tropical areas, and represent popular travel destinations, it appears imperative to accurately assess the risk of infection in order to establish appropriate guidelines for pregnant travellers. 

We launched an international web registry [8] in January 2016 to allow structured collection of data regarding pregnant women and their foetuses exposed to ZIKV. In this article, we present risk assessments for maternal ZIKV infections and adverse pregnancy outcomes among exposed travellers compared to women living in areas with ZIKV circulation using this dataset. 

## 2. Materials and Methods

### 2.1. Study Population and Data Collection

This study utilized the Zika international registry in pregnancy dataset [8]. Health facilities with an antenatal obstetric clinic willing to participate in this international data sharing initiative (available at the time of the study at https://ispso.unige.ch/zika-in-pregnancy-registry/ from 9 March 2017) were invited to systematically enroll all pregnant women attending their clinic that were screened for ZIKV infection at any stage of pregnancy regardless of their infectious status and type of exposure (i.e., exposure through mosquito bites, unprotected sexual intercourse or other). Details regarding participating countries can be found in Annex 1; participating centres have at least one contributing authors in the present paper. All pregnant women exposed to ZIKV at any stage of gestation or prior to gestation were eligible for inclusion in this multicentre study. Exclusion criteria were age <18 years and the inability to consent due to inadequate comprehension of the study purposes. Oral and written information available in English, French, Spanish, Italian and German were provided by the investigators at each centre and oral or written consent obtained. Pregnant women enrolled in the International Zika in Pregnancy registry with an unreported type of exposure or who had not travelled but were exposed through potential sexual transmission were excluded from this analysis. Pregnant women with unreported follow-up after 14 weeks gestation (WG) were also excluded.

Deidentified data were prospectively recorded by each centre using the REDCap (Research Electronic Data Capture) electronic data capture tool [9,10]. Details regarding data collection and validation procedures as well as the collected information can be found in Annex 2. At inclusion (i.e., at the time of ZIKV screening), the following data were recorded: socio-demographic characteristics, obstetrical history and ZIKV exposure. Pregnancies were monitored as clinically indicated according to the local recommendations. After delivery, the following data were collected within 4 weeks: results of maternal testing (ZIKV and/or other infectious pathogens), pregnancy outcomes and neonatal outcomes. 

The study was approved by both the Swiss Ethical Board (CER-VD-2016-00801) and local Ethical boards from the different participating centres. The study was conducted from January 2016 to July 2019. 

### 2.2. Study Group and Exposure Definition

Pregnant women living in areas with ZIKV circulation (residents) were defined as pregnant women whose pregnancy was monitored or who had stayed >6 months in areas where past or active ZIKV circulation had been described according to the Centers for Disease Control and Prevention (CDC) map [7]. Pregnant travellers were defined as pregnant women whose pregnancy was monitored in areas without past or current ZIKV circulation and who had stayed in the above-mentioned areas 6 months. 

### 2.3. Definition of Outcomes

Primary outcome: Absolute risk (%) of maternal ZIKV infection. Exposed women were tested for ZIKV infection according to local recommendations, through serological and molecular testing (RT-PCR). A recent maternal infection was defined by one of the following results: a positive RT-PCR performed either on urine, blood or saliva, or the presence of specific IgM antibodies confirmed by a Plaque reduction neutralization test (PRNT).Secondary outcome: Absolute risk (%) of severe adverse pregnancy outcomes. Foetal and neonatal outcomes were defined as previously described [2,11]. A scoring congenital ZIKV syndrome (CZS) system was created (Appendix A). For multiple gestations, the analysis considered the whole pregnancy. Foetal loss was defined as a spontaneous antepartum foetal death > 14 weeks’ gestation (WG) (i.e., late miscarriages (14–24 WG) and stillbirths (foetal demise >24 WG). Severe adverse pregnancy outcomes were defined as either [1] severely affected foetuses/new-borns and/or [2] foetal loss.

Among exposed foetuses/new-borns, a congenital ZIKV infection was defined either by ZIKV RNA amplification by RT-PCR from at least one foetal/neonatal specimen (placenta, amniotic fluid, cerebrospinal fluid, urine or blood) or identification of ZIKV specific IgM antibodies in the umbilical cord/neonatal blood or in cerebrospinal fluid. 

### 2.4. Statistical Analysis

Absolute risks and the 95% confidence intervals (95% CIs) were estimated using the binomial Wilson score and compared as risk differences (RD) with the relevant 95% CIs. To assess whether travelling was associated with in increased risk of maternal infection, relative risks (RR) were assessed using multivariate Poisson regression models for dichotomous outcomes with robust variance options to estimate the adjusted RR with 95% CIs while controlling for known potential confounding factors and major discrepancies between the study groups. The following variables were included in the model maternal age, maternal comorbidities, aneuploidy and abnormal antenatal screening (defined as an abnormal serology or and non-invasive prenatal testing (NIPT)/amniocentesis). 

Risk factors for maternal infection among pregnant travellers were evaluated in a nested case control study comparing infected pregnant travellers, considered as cases, to non-infected pregnant travellers, taken as controls. Odds ratios were calculated for travelling to South America and the Caribbean Islands compared to other regions (reference group), duration of stay > 2 weeks, > 3 weeks or > 4 weeks compared to those 2 weeks, 3 weeks and 4 weeks, respectively (reference groups) and timing of travel during the epidemic period compared to outside of the epidemic period (reference group). The epidemic period was defined between June 2015 and December 2016, based on the following facts: the first confirmed autochthonous ZIKV case reported in Brazil occurred in early May 2015, the peak of the epidemic in South America occurred during the first half of 2016 [12,13], while in the Caribbean, the epidemic occurred from January 2016 to October 2016 [14]. The end of epidemiological emergency was declared in November 2016. 

To better assess the general impact of each risk factor on the risk of maternal infection, we performed a multivariate analysis. Adjusted odds ratios were adjusted for missing values and for significant risk factors identified in the univariate analysis: travelling during the epidemic (yes/no), dichotomized length of stay and dichotomized region of travel. Except when assessing OR associated with travelling to South America, travelling to the Caribbean Islands was used in the model. Similarly, stays > 2 weeks were used in the model except when assessing longer stays. Collinearity between the variables were assessed using pairwise correlation coefficient. The following relations were assessed: length of stay and travelling during the epidemic, length of stay and travelling to the Caribbean Islands, travelling during the epidemic and travelling to the Caribbean Islands.

Analysis were performed using Stata 14 (Stata Corporation, College Station, TX, USA). A P value inferior of 0.05 was considered as statistically significant. 

Missing values: Maternal comorbidities were considered as negative if not reported, based on the assumption that severe comorbidities are normally well documented. Missing risk of aneuploidy was estimated based on maternal age [15,16]. Based on the hypothesis of missing variables completely at random (MCAR), multiple imputations were performed to increase the power of comparisons and estimate the risks while taking into account missing data on the length of stay, region of travel and period of travel. As significant heterogeneities exist between national standards for prenatal screening, in particular serologies performed during antenatal care, only abnormal serology results were considered.

Sensitivity analysis: We conducted a sensitivity analysis using a broader definition for the diagnosis of a maternal infection: (1) All possible ZIKV infection was defined by one of the following positive results: a positive RT-PCR performed either in urine, blood or saliva, or the presence of specific IgM antibodies confirmed by the PRNT assay and also included pregnant women with only neutralizing antibodies to ZIKV, identified through PRNT assay, without specific IgM antibodies: (2) An active ZIKV infection was defined by a positive RT-PCR performed either in urine, blood or saliva.

## 3. Results

From January 2016 to July 2019, 973 pregnant women were enrolled in the registry and a total of 865 patients were included in the final analysis (Figure 1). Socio-demographic characteristics are presented in Table 1. 

### 3.1. Risk of Maternal ZIKV Infection

#### 3.1.1. Absolute and Relative Risk (RR) of Maternal Infection among Pregnant Travellers Compared to Pregnant Residents

The risk of maternal infection was significantly lower among travellers compared to residents 25.0% (*n* = 36/144) versus 42.9% (*n* = 309/721); crude RR 0.6, 95% CI 0.4–0.8; this remained significant after adjustment for potential confounding factors aRR 0.6, 95% CI 0.4–0.8 (Table 2). Among infected pregnant women, 61.1% (*n* = 22/36) of travellers presented with symptoms compatible with a ZIKV infection compared to 19.4% (*n* = 60/309) of residents (Table 2).

In a sensitivity analysis accounting for different definitions for maternal ZIKV infection, the risk of all possible ZIKV infection remained significantly lower among travellers compared to residents 36.8% (*n* = 53/144) versus 48.1% (*n* = 347/721); RD 11.3%, 95% CI 2.6%–20.0%; crude RR 0.8, 95% CI 0.6–0.9; aRR 0.8, 95% CI 0.6–0.9. When considering active ZIKV infections, there was no difference between travellers compared to residents 16.3% (*n* = 14/86) versus 10.7% (9/84); RD 5.6%, 95% CI 4.7%–15.8%; crude RR 1.5, 95% CI 0.7–3.3; aRR 1.4, 95% CI 0.6–3.3). This subgroup was small, given RT-PCR results were only available for 11.6% of residents (*n* = 84/721) compared to 59.7% of travellers (*n* = 86/144).

#### 3.1.2. Risk Factors for Maternal Infection among Pregnant Travelers

We performed a nested case control study to evaluate potential risk factors for maternal infections (Table 3). Travelling during the epidemic [crude OR 46.4, 95% CI 7.0–1916.5] and travelling to the Caribbean islands compared to other regions [crude OR 5.0, 95% CI 2.0–12.6] were associated with an increased risk of maternal infection. Similarly, a duration of stay >2 weeks [crude OR 12.8, 95% CI 1.9–541.3] and a duration of stay > 3 weeks [crude OR 2.9, 95% CI 1.0–8.9] compared to those ≤ 2 weeks or ≤ 3 weeks, respectively, were both associated with an increased risk for maternal infection. Similar findings were also observed when considering all possible ZIKV infections or active ZIKV infections, except that a duration of stay > 4 weeks compared to those ≤ 4 weeks was also associated with an increased risk of possible ZIKV infections.

In a multivariate analysis accounting for missing values through multiple imputation, travelling during the epidemic period, travelling to the Caribbean Islands and a duration of stay > 2 weeks were independently associated with the risk of maternal infection [aOR 29.4, 95% CI 3.7–228.1 for travelling during the epidemic period, aOR 3.2, 95% CI 1.2–8.7 for travelling to the Caribbean Islands and aOR 8.7, 95% CI 1.1–71.5 for stays abroad >2 weeks, respectively] when compared to travelling outside of the epidemic, to other regions or stays ≤ 2 weeks, respectively (Table 3). In contrast, a duration of stay >3 weeks compared to ≤ 3 weeks was not associated with a significant increased risk of maternal infection [aOR 1.5, 95% CI 0.5–4.5] (Table 3). When considering all possible ZIKV infection, these associations remained significant. In addition, a duration of stay >3 weeks was also associated with an increased risk of possible ZIKV maternal infections when compared to ≤3 weeks [aOR 3.5, 95% CI 1.2–10.0], but a duration of stay >4 weeks was not associated with an increased risk compared to ≤ 4 weeks [aOR 1.9, 95% CI 0.7–5.1]. Active ZIKV infections were not tested because the sample size was considered too limited.

### 3.2. Risk of Adverse Pregnancy Outcomes

#### 3.2.1. Absolute and Relative Risk for Adverse Pregnancy Outcomes among Exposed Pregnant Travellers Compared to Pregnant Residents

Overall, the risk of severe adverse pregnancy outcomes within the cohort was 7.8% (*n* = 67/865), including 8.3% (*n* = 3/36) among travellers and 12.7% (*n* = 39/309) among residents. (Table 4). Asymptomatic new-borns were more frequently observed among travellers with a recent maternal ZIKV infection compared to infected residents [91.7% (*n* = 33/36) *versus* 76.7% (*n* = 237/309)] (Table 4). 

Results of foetal/neonatal testing were available in 20.1% (*n* = 29/144) of travellers and 87.8% residents (*n* = 633/721). A congenital infection was confirmed in 17.2% [(*n* = 5/29), 95% CI 7.6%–34.5%)] of foetuses/new-borns among travellers with available testing and 12.2% [(*n* = 77/633), 95% CI 9.8%–14.9%] among residents (Appendix A). Interestingly, one confirmed congenital infection among residents occurred in a woman with negative ZIKV testing. This woman was identified to be IgG positive with positive PRNT testing and was therefore considered as a possible ZIKV infection. The new-born was asymptomatic at birth and specific IgM antibodies were detected. Among infected pregnant women, materno-foetal transmission rate was 21.7% (*n* = 5/23) among travellers and 25.9% (*n* = 76/293) among residents (Table 4). 

#### 3.2.2. Absolute Risk of Adverse Pregnancy Outcomes among Infected Travellers Compared to Non-Infected Pregnant Travellers

Adverse pregnancy outcomes were observed in 8.3% (*n* = 3/36) of infected travellers *versus* 3.7% (4/108) among non-infected travellers. Findings of cases with severe adverse pregnancy outcomes among travellers are presented in Table 5. Of five cases with a confirmed foetal infection (Appendix A), three cases had severe adverse pregnancy outcomes. All women with severe adverse pregnancy outcomes were exposed during the first trimester of pregnancy and had travelled more than 2 weeks, during the recent epidemic; of note, two patients experienced symptoms. Among negative mothers, four severe adverse pregnancy outcomes were recorded, one of which occurred in a mother with a *possible ZIKV infection* (Appendix A). The new-born presented with isolated macular anomalies; he was unfortunately not tested for ZIKV infection.

## 4. Discussion

We present here the first prospective study assessing the risks of maternal ZIKV infection and adverse pregnancy outcomes among travellers compared to residents. The absolute risk of maternal infection was significantly lower for travellers, with a 25% absolute risk over the study period. Importantly, the risk of maternal infection was related to the presence of an ongoing outbreak and the length of stay abroad, as well as the region of travel. Although the numbers were low limiting the generalization of the result, when considering only pregnant travellers that travelled outside the epidemic period or less than 2 weeks, the risk was reduced to 1.7% (1/60) and 3.2% (1/31), respectively. No maternal infections were recorded among pregnant travellers outside the epidemic period and with a length of stay abroad less than two weeks. 

Two studies performed in Spain in 2016-2017 observed an incidence of recent/confirmed maternal infection of 1.3% (14/1057) [17] and 3.5% (9/254), respectively [18], while during the 2009–2018 period, Norman et al. observed a 3.8% incidence of arboviral infections among 861 returning travellers, of which 12% were caused by ZIKV [19]. The higher proportion of maternal infection in our study might be related to the inclusion of a majority of women exposed during the recent epidemic and a high detection rate, as all patients were tested. Interestingly, Norman et al. found no association with the length of stay. In their study, most patients had a length of stay > 2 weeks with a median length of stay of 23 days (interquartile range 15 to 55 days) [19]. Travelling to the Caribbean Islands was associated with an increased risk of maternal infection. This association might be explained by the relative homogeneity of the Caribbean region in terms of factors contributing to the cycle of ZIKV vectoral transmission (i.e., climate and Aedes spp. distribution, population densities) compared to South America, where significant socio-ecological variations are observed. As such, the incidence of ZIKV infections in Brazil between 2015–2016 was highly variable depending on the region, with the southern parts of the country, including urban areas of Sao Paulo, being spared [1]. 

We observed an incidence of severe adverse pregnancy outcomes similar to what has been reported previously. In the US territories, the incidence of severe foetal/neonatal anomalies observed among infected patients ranged from 4% to 8% depending on the gestational trimester of suspected maternal infection [3], while in the Caribbean region, severe adverse outcomes were reported in 8.1% of foetuses [20]. This highlights that the majority of exposed foetuses (>90%) will remain asymptomatic or pauci-symptomatic, even in the case of a confirmed foetal infection [2]. 

Our study has limitations. First, the diagnosis of a recent ZIKV infection is challenging. We used criteria based on both nucleic acid amplification testing (NAAT) and serology. NAAT are limited by the transient character of the ZIKV viremia [21]. On the other hand, serology is poorly reliable, especially in secondary flavivirus infections. Re-infections are associated with cross-reactions of both specific IgM and neutralizing antibodies. Moreover, secondary stimulations may suppress the production of specific antibodies [22]. In that context, a negative IgM testing does not necessarily exclude a recent infection, as observed in two of our cases, in which congenital ZIKV infection was confirmed by the identification of specific IgM in one of the new-borns, while in the other, severe macular anomalies compatible with ZIKV were observed. To overcome this limitation, we performed a sensitivity analysis including different definitions of exposure, possible ZIKV infections versus active infections (positive viremia). The first strategy supported our findings, while risk estimates using the active infection definition lost their statistical significance owing to the small sample size. These aspects further highlight the difficulties related to the diagnosis of ZIKV infection in pregnant women and further argue against routine screening of exposed women [1]. 

Second, our study is limited by the small number of cases with adverse pregnancy outcomes among travellers. Though, it allowed us to correctly assess risk factors for maternal infection, our study was not powered to detect differences in pregnancy outcomes between infected and non-infected pregnant travellers and to assess potential contributing factors (e.g., timing of maternal infection, persistent viremia). Furthermore, our study did not assess long term outcomes among new-borns, which may lead to an underestimation of the consequences of congenital infection. In addition, we were not able to capture miscarriages in a systematic way. To avoid underreporting or misclassification, we excluded all pregnant women with unreported outcomes after 14WG. This might have underestimated the rate of adverse outcomes related to ZIKV infection in early pregnancy. Exact rates of miscarriage are difficult to assess, due to the high frequency of unreported early-stage pregnancy loss and might be as high as 30 to 40 % [23]. As to whether maternal ZIKV infection increases this risk remains unclear. 

Finally, our study is based on a registry and not systematic sampling. As such, we observed an overrepresentation of symptomatic women among travellers, as asymptomatic women may not have sought medical care. Nevertheless, as symptoms have not been correlated to worse foetal outcomes, the impact of this bias on our results seems limited. Furthermore, although we develop a user-friendly system to collect data in a systematic way, as with all observational studies missing data are inevitable. To account for this we performed multiple imputations, allowing us an acceptable evaluation. 

Our study focused on pregnant women. Nevertheless, we believe that our conclusions may be extended to young couples trying to conceive. Although, we did not assess the risks associated with sexual transmission, the probability for a male to subsequently infect his partner is related to his initial risk of infection. Several agencies, including the WHO, CDC and National Travel Health Network and Centre (NaTHNaC) recommend waiting 2 months for women and 3 months for men before getting pregnant after travelling to areas with both current and past outbreaks [6,7,24]. These recommendations might be overly cautious. Based on the present finding of relatively low risk, it seems reasonable not to advise any delay for patients travelling to areas without any current outbreaks who are staying 2 weeks; as supported by the Swiss public health institute [25]. Precaution to avoid mosquitoes bites should nevertheless be strictly applied. Furthermore, recommendations for travelling pregnant women should also take into additional exposure to other infections pathogens. Most areas with ZIKV circulation are also endemic for DENV, CHIKV or malaria. Growing evidence suggests a negative impact of DENV and CHIKV on pregnant women and their offspring [26,27], while Malaria remains a major cause of stillbirth in endemic countries [28].

## 5. Conclusions

We provided a reliable assessment of the risks of maternal ZIKV infection and associated risk of adverse pregnancy outcomes among travellers. Our findings suggest the risk of maternal infection among travellers is lower to what is observed for pregnant residents. The specific risk of maternal infection for travellers is related to the presence of ongoing outbreaks and stay duration, with stays < 2 weeks associated with a low risk in the absence of ongoing outbreaks. 

## Figures and Tables

**Figure 1 viruses-13-00341-f001:**
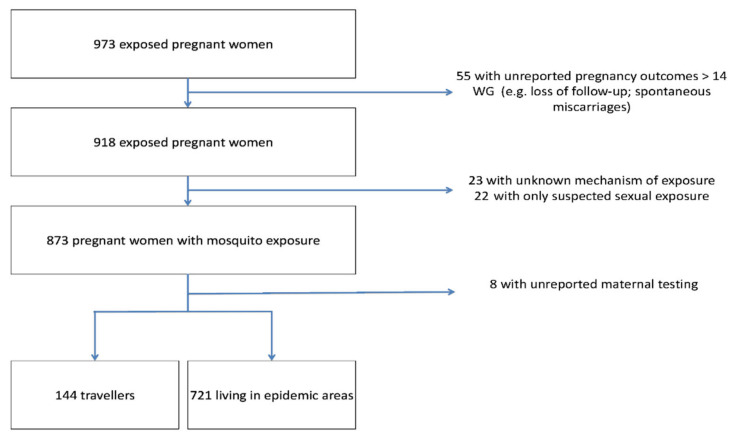
Flow chart. Abbreviations: WG, weeks’ gestation; ZIKV, Zika virus.

**Table 1 viruses-13-00341-t001:** Maternal characteristics within the cohort. Abbreviations: DS, Down syndrome; HTD, Hypertensive disorders; IQR, interquartile range; NIPT, Non-Invasive Prenatal Testing; y.o., years old; ZIKV, Zika Virus.

Socio-Demographic Factors			Pregnant Travellers			Pregnant Women Living in Endemic Areas		
			*n* = 144			*n* = 721		
			All Pregnant Travellers	Recent Maternal ZIKV Infection	Negative ZIKV Infection	All Pregnant Women Living in Endemic Areas	Recent Maternal ZIKV Infection	Negative ZIKV Infection
			*n* = 144	*n* = 36	*n* = 108	*n* = 721	*n* = 309	*n* = 412
**Maternal age**								
	Median—y.o. (IQR)		31 (27–35)	28.5 (24–31.5)	32 (28–35)	28 (23–34)	27.1 (22.4–32.7)	28.3 (23.6–34.0)
	Age > 35 y.o.—no (%)		38 (26.4)	5 (13.9)	33 (30.6)	193 (26.8)	52 (16.8)	141 (34.2)
**Ethnicity**								
	Caucasian		29 (20.1)	2 (5.6)	27 (25.0)	1 (0.1)	0 (0.0)	1 (0.2)
	Hispanic or latino-american		47 (32.6)	16 (44.4)	31 (28.7)	644 (89.3)	300 (97.1)	344 (83.5)
	Afro-american		5 (3.5)	0 (0.0)	5 (4.6)	2 (0.3)	0 (0.0)	2 (0.5)
	Asian or Pacific Islands		8 (5.6)	1 (2.8)	7 (6.5)	2 (0.3)	0 (0.0)	2 (0.5)
	Other		3 (2.1)	0 (0.0)	3 (2.8)	0 (0.0)	0 (0.0)	0 (0.0)
	Unknown		52 (36.1)	17 (47.2)	35 (32.4)	72 (10.0)	9 (2.9)	63 (15.3)
**Previous pregnancies—no (IQR)**								
	Nulliparous—no (%)		76 (52.8)	18 (50.0)	58 (53.7)	307 (42.6)	105 (34.0)	202 (49.0)
	Multiparous		68 (47.2)	18 (50.0)	50 (46.3)	414 (57.4)	204 (66.0)	210 (51.0)
	Multiparous ≥ 3		5 (3.5)	2 (5.6)	3 (2.8)	296 (41.1)	141 (45.6)	155 (37.6)
**Previous adverse pregnancy outcomes—no (%)**								
	Stillbirths		11 (7.6)	0 (0.0)	11 (10.2)	27 (3.8)	10 (3.2)	17 (4.1)
	Spontaneous abortions		44 (30.6)	11 (30.6)	33 (30.6)	147 (20.4)	45 (14.6)	102 (24.8)
**Maternal comorbidities—no (%)**								
	All maternal comobidities		50 (34.7)	10 (27.8)	40 (37.0)	292 (40.5)	125 (40.4)	167 (40.5)
	Diabetes (previous or gestational)							
		Previous	1 (0.7)	0 (0.0)	1 (0.9)	7 (1)	3 (1.0)	4 (1.0)
		Gestational	6 (4.2)	1 (2.8)	5 (4.6)	29 (4.0)	11 (3.6)	18 (4.4)
		unknown	45 (31.3)	20 (55.6)	25 (23.1)	80 (11.1)	17 (5.5)	63 (1.5)
	Thyroid dysfunction							
		Hypothyroidism	7 (4.9)	1 (2.8)	6 (5.6)	3 (0.4)	1 (0.3)	2 (0.5)
		Hyperthyroidism	0 (0.0)	0 (0.0)	0 (0.0)	0 (0.0)	0 (0.0)	0 (0.0)
		Unknown	56 (38.9)	20 (55.6)	36 (33.3)	693 (96.1)	293 (94.8)	400 (97.1)
	Vascular pathologies							
		Pre-existing HTA	1 (0.7)	1 (2.8)	0 (0.0)	9 (1.3)	5 (1.6)	4 (1.0)
		Gestational/pre-eclampsia	3 (2.1)	0 (0.0)	7 (6.5)	34 (4.7)	17 (5.5)	17 (4.1)
		Unknown	15 (10.4)	10 (27.8)	5 (4.6)	12 (1.7)	10 (3.2)	2 (0.5)
	Drugs							
		Cigarettes	6 (4.2)	0 (0.0)	6 (5.6)	1 (0.1)	0 (0.0)	1 (0.2)
		Alcool	5 (3.5)	0 (0.0)	5 (4.6)	12 (1.7)	6 (1.9)	6 (1.5)
		Unknown	48 (33.3)	20 (55.6)	28 (25.9)	81 (11.2)	18 (5.8)	63 (1.5)
**Antenatal screening**								
	Aneuploidy screening							
		Risk of T21 > 1/1000	6 (4.2)	1 (2.8)	5 (4.6)	108 (15.0)	26 (8.4)	82 (19.9)
		Unknown T21 risk	82 (56.8)	29 (80.6)	53 (49.1)	273 (37.9)	146 (47.3)	127 (30.8)
	Genetic screening (NIPT/Amniocentesis)							
		Abnormal	1 (0.7)	0 (0)	1 (0.9)	1 (0.1)	1 (0.3)	0 (0)
	Serologies screening							
		Abnormal	0 (0)	0 (0)	0 (0)	3 (0.4)	0 (0)	3 (0.7)

**Table 2 viruses-13-00341-t002:** Risk of maternal infection. Abbreviations: aRR, adjusted risk ratio; CI; Confidence interval; RD, Risk difference; RR, Risk ratio; ZIKV, Zika virus. * adjusted for missing values length of stay, region of travel and travelling during the epidemic.

	Pregnant Travellers		Pregnant Residents						
	*n* = 144		*n* = 721						
	*n* (%)	95% CI	*n* (%)	95% CI	RD (95% CI)	Crude RR (95% CI)	*p* Value	aRR (95% CI)	*p* Value
Maternal infection									
Recent Maternal infection	36 (25.0)	18.2–32.9	309 (42.9)	39.2–46.6	17.9 (25.8–1.0)	0.6 (0.4–0.8)	0.0001	0.6 (0.4–0.8)	0.0001
Symptomatic infection	22 (61.1)	43.5–76.9	60 (19.4)	15.2–24.3	41.7 (25.2–58.2)	3.1 (2.2–4.4)	<0.0001	3.0 (2.1–4.3)	<0.0001

* adjusted for maternal age (>35 y.o. cat), maternal comorbidities (yes/no), risk of aneuploidy (yes/no) and abnormal prenatal screening (yes/no).

**Table 3 viruses-13-00341-t003:** Risk factors for maternal infection among pregnant travellers. Abbreviations: aOR, adjusted odds ratio; CI, Confidence interval; OR, Odds Ratio; ZIKV, Zika virus; n.a., not applicable; * adjusted for missing values “length of stay”, “region of travel” and “travelling during the epidemic”.

Exposition			Pregnant Travellers							
			Recent Maternal ZIKV Infection *n* = 36		Negative Maternal ZIKV Infection *n* = 108					
			*n* (%)	95% CI	*n* (%)	95% CI	Crude OR (95% CI)	*p* Value	aOR (95% CI)	*p* Value
Region of travelling										
	Known		35 (97.2)	85.8–99.5	102 (94.4)	88.4–97.4				
		South America	11 (31.4)	18.6–48.0	46 (45.1)	35.8–54.8	0.6 (0.2–1.4)	0.1705	0.4 (0.2–1.1)	0.087
		Carribean	24 (68.6)	52.0–81.4	31 (30.4)	22.3–39.8	5.0 (2.0–12.6)	0.0001	3.2 (1.2–8.7)	0.023
		South East Asia	0 (0)	0–9.8	12 (11.8)	6.9–19.4				
		Africa	0 (0)	0–9.8	7 (6.9)	3.4–13.5				
		Other	0 (0)	0–9.8	6 (5.9)	2.7–12.2				
	Unknown		1 (2.8)	0.5–14.2	6 (5.6)	2.6–11.6				
Length of stay										
	Known		27 (75.0)	58.9–86.2	91 (84.3)	76.3–89.9				
		≤2 weeks	1 (3.7)	6.5–18.3	30 (33.0)	24.2–43.1	12.8 (1.9–541.3)	0.0021	8.7 (1.1–71.5)	0.0041
		≤3 weeks	7 (25.9)	13.2–44.7	46 (50.6)	40.5–60.6	2.9 (1.0–8.9)	0.0284	1.5 (0.5–4.5)	0.471
		≤4 weeks	12 (44.4)	27.6–62.7	51 (56.0)	45.8–65.8	1.6 (0.6–4.2)	0.3801	0.8 (0.3–2.3)	0.752
		>4 weeks	15 (55.6)	37.3–72.4	40 (44.0)	34.2–54.2				
		Unknown	9 (25.0)	13.8–41.0	17 (15.7)	10.1–23.8				
Period of exposure										
	Known		34 (94.4)	81.9–98.5	101 (93.5)	87.2–96.8				
	During the epidemic peak		33 (97.1)	85.1–99.5	42 (41.6)	32.5–51.3	46.4 (7.0–1916.5)	<0.0001	29.4 (3.7–228.1)	0.001
		July–Dec 2015	1 (3.0)	0.5–15.3	1 (2.3)	0.4–12.3				
		Jan–June 2016	25 (75.8)	59.0–87.2	20 (47.6)	33.4–62.3				
		July–Dec 2016	7 (21.2)	10.7–37.8	21 (50.0)	35.5–64.5				
	Outside of the epidemic peak		1 (2.9)	0.5–14.9	59 (58.4)	48.7–67.5				
		Prior to June 2015	0 (0)	n.a.	1 (1.7)	0.3–9.0				
		2017	1 (100)	n.a.	55 (93.2)	83.8–97.3				
		2018	0 (0)	n.a.	3 (5.1)	1.7–13.9				
	Unknown		2 (5.6)	1.5–18.1	7 (6.5)	3.2–12.8				
Use of mosquitoes’ repellent										
	Known		10 (27.8)	15.8–44.0	52 (48.1)	39.0–57.5				
		Use of repellent	8 (80.0)	49.0–94.3	28 (53.8)	40.5–66.6	3.5 (0.6–35.5)	0.1705	1.1 (0.1–7.9)	0.925
	Unknown		26 (72.2)	56.0–84.2	56 (51.9)	42.5–61.0				

* adjusted for missing values length of stay, region of travel, and travelling during the epidemic.

**Table 4 viruses-13-00341-t004:** Adverse pregnancy outcomes among infected pregnant travellers compared to infected pregnant residents. Abbreviations: CI; Confidence interval; ZIKV, Zika virus.

		Positive Recent Maternal ZIKV Infection			
		Pregnant Travellers		Pregnant Residents	
		*n* = 36		*n* = 309	
		*n* (%)	95% CI	*n* (%)	95% CI
**Foetal/Neonatal outcomes**					
	Asymptomatic	33 (91.7)	78.2–97.1	237 (76.7)	71.7–81.1
	Severe adverse pregnancy outcomes	3 (8.3)	2.9–21.8	39 (12.6)	9.4–16.8
**Foetal/neonatal testing**					
Known		23 (63.9)	47.6–77.5	293 (94.8)	91.7–96.8
	Positive	5 (21.7)	9.7–41.0	76 (25.9)	21.3–31.2
	Negative	18 (78.3)	58.1–90.3	217 (70.2)	64.9–75.1
Unknown		13 (36.1)	22.5–52.4	16 (5.2)	3.2–8.2

**Table 5 viruses-13-00341-t005:** Adverse pregnancy outcomes among infected pregnant travellers compared to non-infected pregnant travellers within a nested case-control study. The risk of severe adverse pregnancy outcomes associated with a recent maternal ZIKV infection among pregnant travellers was evaluated in a nested case control study comparing infected pregnant travellers, considered as cases, to non-infected pregnant travellers, taken as controls. Abbreviations: ZIKV, Zika virus.

		Travellers			
		Positive Recent Maternal ZIKV Infection		Negative Recent Maternal ZIKV Infection	
		*n* = 36		*n* = 108	
		*n* (%)	95% CI	*n* (%)	95% CI
**Foetal/Neonatal outcomes**					
	Asymptomatic	33 (91.7)	78.2–97.1	103 (95.4)	89.6–98.0
	Severe adverse pregnancy outcomes	3 (8.3)	2.9–21.8	4 (3.7)	1.4–9.1
**Foetal/neonatal testing**					
Known		23 (63.9)	47.6–77.5	6 (5.6)	2.6–11.6
	Positive	5 (21.7)	9.7–41.0	0 (0.0)	n.a.
	Negative	18 (78.3)	58.1–90.3	6 (100.0)	61.0–100.0
Unknown		13 (36.1)	22.5–52.4	102 (94.4)	88.4–97.4

## Data Availability

The data presented in this study are available on request from the corresponding author.

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
