# Peer review of "Maternal Infection and Adverse Pregnancy Outcomes among Pregnant Travellers: Results of the International Zika Virus in Pregnancy Registry"

_viruses, 2021, doi:10.3390/v13020341_

Round 1

Reviewer 1 Report

The study compares the rates of Zika virus infection in travellers and residents and describes several cases of fetal/congenital adverse events associated with the infection, based on an international registry. It is well written and special effort has been done in evaluating different variables to explore potential risk factors.

The finding that the risk for pregnant residents is higher than for travellers may seem obvious due to longer exposure to infected mosquitoes, but it is certainly valuable to be confirmed.  Although it might be expected, the fact that pregnant women travelling to areas outside of the epidemic are at low risk of Zika virus infection is also relevant. This could contribute to update guideliness for travellers and to balance cost effectiveness of screening programs.

Specific comments:

1) Lines 72-74: the web site indicated (https://ispso.unige.ch/zika-in-pregnancy-registry/) does not seem to work and the reference 8 does not provide information on the Zika international registry in pregnancy dataset. Further information about this initiative is therefore unclear for the reader.

2) Information is missing regarding the local guidelines used for testing in each of the centers including the patients. It should be specified how many patients were recruited in each of the participating centers, and under which testing guidelines the pregnant women were screened. Different testing algorithms have been used in different countries and this could result in different detection rates of infected patients.

3) Laboratory diagnosis. Recent maternal infection was defined by a positive RT-PCR test or a positive IgM + PRNT.

  • The authors should indicate how many patients in each group were diagnosed by RT-PCR and by serological methods. Serological diagnosis can be challenging specially in secondary flavivirus infections and people previously vaccinated. For example, the certainty provided by serological diagnosis can be significantly different in an unvaccinated traveller that has never visited dengue endemic areas than in a resident vaccinated against yellow fever that has experienced one or more episodes of dengue. RT-PCR detection, in contrast, provides a diagnosis of confirmation in both cases.

  • Laboratory methods are crucial in arbovirus diagnosis. There is a lack of information that should be addressed indicating the references or commercial assays used for both RT-PCR and antibody detection.

  • Which was the cut off for the PRNT method to confirm the presence of neutralizing antibodies?

  • Were the patients tested for other arboviruses? Was serology (IgM/IgG/PRNT) performed against other possible circulating flaviviruses such as dengue, west nile or yellow fever? It seems especially important to know the data on dengue serology, as it might be a very common disease in the Zika endemic areas described, and Zika and dengue antibodies are known to cross-react in a variety of assays.

Author Response

Reviewer 1

The study compares the rates of Zika virus infection in travellers and residents and describes several cases of fetal/congenital adverse events associated with the infection, based on an international registry. It is well written and special effort has been done in evaluating different variables to explore potential risk factors.

The finding that the risk for pregnant residents is higher than for travellers may seem obvious due to longer exposure to infected mosquitoes, but it is certainly valuable to be confirmed.  Although it might be expected, the fact that pregnant women travelling to areas outside of the epidemic are at low risk of Zika virus infection is also relevant. This could contribute to update guidelines for travellers and to balance cost effectiveness of screening programs.

We thank the reviewer for this positive comment.

Specific comments:

1) Lines 72-74: the web site indicated (https://ispso.unige.ch/zika-in-pregnancy-registry/) does not seem to work and the reference 8 does not provide information on the Zika international registry in pregnancy dataset. Further information about this initiative is therefore unclear for the reader.

The website has now been closed as the study was closed. To acknowledge this aspect, we added the following comment in the method section: “Health facilities with an antenatal obstetric clinic willing to participate in this international data sharing initiative (available at the time of the study at https://ispso.unige.ch/zika-in-pregnancy-registry/) were invited to systematically enrol all pregnant women attending their clinic”.

As suggested by the reviewer, we added an annex (Annex 2) detailing the procedures and quality control used in the Zika in pregnancy Registry as well as a table with the collected information for each enrolled woman. The method section was modified as follows:

Lines (84-85) “Details regarding participating countries can be found in Annex 1; participating centres have at least one contributing authors in the present paper”

Lines (105-106) “Deidentified data were prospectively recorded by each centre using the REDCap (Research Electronic Data Capture) electronic data capture tool (9,10). Details regarding data collection and validation procedures as well as the collected information can be found in Annex 2.”

2) Information is missing regarding the local guidelines used for testing in each of the centers including the patients. It should be specified how many patients were recruited in each of the participating centers, and under which testing guidelines the pregnant women were screened. Different testing algorithms have been used in different countries and this could result in different detection rates of infected patients.

Screening policies varied depending on countries and timing of the epidemic. Some endemic countries (e.g. French Guinea) offered a systematic screening of all pregnant women during the epidemic period, by opposition most policies in non-endemic countries offered screening in case of compatible symptoms. As the local policies gradually evolved during the epidemic, we cannot provide one unique policy. We agree that this could have led to differences in term of proportions of symptomatic infections observed. This is acknowledged in the discussion

“Third, we observed an overrepresentation of symptomatic women among travellers, as asymptomatic women may not have sought medical care. Nevertheless, as symptoms have not been correlated to worse foetal outcomes, the impact of this bias on our results seems limited.”

Nevertheless, as suggested by the reviewer and for transparency purpose, we added a supplementary table providing a list of all participating centres and the number of patients included in each centre (Annex 1).

3) Laboratory diagnosis. Recent maternal infection was defined by a positive RT-PCR test or a positive IgM + PRNT.

  • The authors should indicate how many patients in each group were diagnosed by RT-PCR and by serological methods. Serological diagnosis can be challenging specially in secondary flavivirus infections and people previously vaccinated. For example, the certainty provided by serological diagnosis can be significantly different in an unvaccinated traveller that has never visited dengue endemic areas than in a resident vaccinated against yellow fever that has experienced one or more episodes of dengue. RT-PCR detection, in contrast, provides a diagnosis of confirmation in both cases.

We agree that a diagnosis based only RT-PCR, by opposition to serological diagnosis, is associated with a low rate of false positive results. Nevertheless, false negative results may often be observed especially in asymptomatic infections, as viremia is transitory. Therefore, serological diagnosis is often preferred in asymptomatic patients. In that context, we decided to also consider the specific detection of IgM in our diagnostic criteria as long as they were confirmed by PRNT. Nevertheless, we agree with the reviewer that serological diagnosis can be challenging especially in secondary infection. As such, we performed a sensitivity analysis accounting for different diagnostic criteria:

Methods (lines 178-184) : “Sensitivity analysis: We conducted a sensitivity analysis using a broader definition for the diagnosis of a maternal infection : (1) All possible ZIKV infection was defined by one of the following positive results: a positive RT-PCR performed either in urine, blood or saliva, or the presence of specific IgM antibodies confirmed by the PRNT assay and also included pregnant women with only neutralizing antibodies to ZIKV, identified through PRNT assay, without specific IgM antibodies ; (2) An active ZIKV infection was defined by a positive RT-PCR performed either in urine, blood or saliva.”

The number of patients tested with RT-PCR is mentioned in the result section (lines 206-208):

“This subgroup was small, given RT-PCR results were only available for 11.6% of residents (n=84/721) compared to 59.7% of travellers (n=86/144) (data not shown).”

  • Laboratory methods are crucial in arbovirus diagnosis. There is a lack of information that should be addressed indicating the references or commercial assays used for both RT-PCR and antibody detection.

We agree with the reviewer’s comment that references of commercial assays are crucial. Unfortunately, as this study was conducted over a large period of time (i.e. January 2016 – July 2019) in several centres, specific commercial assays have evolved depending on the local capacities. Nevertheless, all centres used assays that had been previously validated by their national reference centres, insuring their reliability.

  • Which was the cut off for the PRNT method to confirm the presence of neutralizing antibodies?

 A 90% neutralization was used as a cut off for positivity (i.e. PRNT90).

  • Were the patients tested for other arboviruses? Was serology (IgM/IgG/PRNT) performed against other possible circulating flaviviruses such as dengue, west nile or yellow fever? It seems especially important to know the data on dengue serology, as it might be a very common disease in the Zika endemic areas described, and Zika and dengue antibodies are known to cross-react in a variety of assays.

We agree with the reviewer’s comment. In endemic countries, centres also tested for other Flavivirus as suggested by Arboviral references centres.

Limitations regarding the risk of cross-reactions is acknowledged in the discussion (lines 328-341):

On the other hand, serology is poorly reliable, especially in secondary flavivirus infections. Re-infections are associated with cross-reactions of both specific IgM and neutralizing antibodies. Moreover, secondary stimulations may suppress the production of specific antibodies (19). In that context, a negative IgM testing does not necessarily exclude a recent infection, as observed in two of our cases, in which congenital ZIKV infection was confirmed by the identification of specific IgM in one of the new-borns, while in the other, severe macular anomalies compatible with ZIKV were observed. To overcome this limitation, we performed a sensitivity analysis including different definitions of exposure, possible ZIKV infections versus active infections (positive viremia). The first strategy supported our findings, while risk estimates using the active infection definition lost their statistical significance owing to the small sample size. These aspects further highlight the difficulties related to the diagnosis of ZIKV infection in pregnant women and further argue against routine screening of exposed women (1).”

Of note, a significant proportion of our patients from endemic countries came from French Guinea (see Annex 1). In this country, circulation of other Flaviviruses was expected to be minimal at the time of the study as circulation of Dengue virus has been very low since 2014 (Flamand, Euro Surveill, 2017).

Reviewer 2 Report

In this manuscript, the authors presented a study that showed the risks of maternal ZIKV infection and adverse pregnancy outcomes among travelers compared to residents. The findings suggest that the risk of maternal infection among travelers was lower compared to what was observed for pregnant residents. This paper provided a reliable assessment of the risks of maternal ZIKV infection and the associated risk of adverse pregnancy outcomes among travelers.

Minor comment:

There is a lot of abbreviations used in the manuscript which makes it difficult to read. It would be better if there is an abbreviation list that also includes a short explanation for each abbreviation.

Author Response

Reviewer 2

Comments and Suggestions for Authors

In this manuscript, the authors presented a study that showed the risks of maternal ZIKV infection and adverse pregnancy outcomes among travelers compared to residents. The findings suggest that the risk of maternal infection among travelers was lower compared to what was observed for pregnant residents. This paper provided a reliable assessment of the risks of maternal ZIKV infection and the associated risk of adverse pregnancy outcomes among travelers.

 We thank the reviewer for this positive comment.

Minor comment:

There is a lot of abbreviations used in the manuscript which makes it difficult to read. It would be better if there is an abbreviation list that also includes a short explanation for each abbreviation.

As suggested a list of abbreviations was added.

Reviewer 3 Report

This is an interesting study about Zika infection among pregnancy and their outcomes. The authors did extensive analysis but I have some queries about your study and need some revisions.

  1. How do you validate the data from web registry? Please describe about the validation procedure for your data. How many countries involve in this study and who did the data submission and how to check?
  2. The authors also described they took consent from the study participants and they also described the data from international web registry. If the authors took consent please clearly describe how to take consent (either written or verbal) for enrolling in this study. Who take the consent during this study. We are not clear the study flow and procedure. Please rewrite to get clear understanding to the authors.
  3. The country origin of the pregnant mother is very important as many flaviviruses (Dengue, JEV, West Nile, etc) are co-circulating. Therefore, it is difficult to confirm Zika infection in some cases by serological tests only. How do you classify those cases. Therefore, it is important to express the country list at method section whether it is endemic for other flaviviruses or not.
  4. Please describe the definition of possible zika virus infection.
  5. The data for showing risks of resident mother is a little bit awarding. The proportion of ZKV infection among resident other is high compared to traveler. The problem is the denominator of the pregnant other for resident. The authors took denominator as the number of pregnant women enrolled in this study. Is it possible to conclude like this?

Author Response

Reviewer 3

Comments and Suggestions for Authors

This is an interesting study about Zika infection among pregnancy and their outcomes. The authors did extensive analysis but I have some queries about your study and need some revisions.

  1. How do you validate the data from web registry? Please describe about the validation procedure for your data. How many countries involve in this study and who did the data submission and how to check?

As suggested by reviewer 1, we added a Table (Annex 1) with the patients recruited in each centre. In addition to better clarify data collection procedures and quality checks, we added an annex (Annex 2) detailing the procedures and quality control used in the Zika in pregnancy Registry (See answer to Reviewer 1).

  1. The authors also described they took consent from the study participants and they also described the data from international web registry. If the authors took consent please clearly describe how to take consent (either written or verbal) for enrolling in this study. Who take the consent during this study. We are not clear the study flow and procedure. Please rewrite to get clear understanding to the authors.

Both oral and written information were provided by the investigators at each centre and both oral and written consent were accepted. For written information, an information sheet as well as a consent form were available in French, English, Spanish, Italian and German. If required by the reviewer, we would be happy to provide him these documents.

To further clarify these aspects, we added the following sentence in the method section: “Oral and written information available in English, French, Spanish, Italian and German were provided by the investigators at each centre and oral or written consent obtained.”

  1. The country origin of the pregnant mother is very important as many flaviviruses (Dengue, JEV, West Nile, etc) are co-circulating. Therefore, it is difficult to confirm Zika infection in some cases by serological tests only. How do you classify those cases. Therefore, it is important to express the country list at method section whether it is endemic for other flaviviruses or not.

We agree with reviewer’s comment that serological diagnosis can be challenging especially in secondary infections. Please see our answer to Reviewer 1 (last point).

Furthermore, as previously suggested by Reviewer 1, we added a list of participating countries and the numbers of patients included. Maternal origin and region of residence are described in Table 1, and region of travel in Table 3. 

  1. Please describe the definition of possible zika virus infection.

The possible ZIKV infection definition includes both all possible and confirmed infections (positive RT-PCR performed either in urine, blood or saliva). The specific definition is explained in the methods section (sensitivity analysis): “(1) All possible ZIKV infection was defined by one of the following positive results: a positive RT-PCR performed either in urine, blood or saliva, or the presence of specific IgM antibodies confirmed by the PRNT assay and also included pregnant women with only neutralizing antibodies to ZIKV, identified through PRNT assay, without specific IgM antibodies”

  1. The data for showing risks of resident mother is a little bit awarding. The proportion of ZKV infection among resident mother? is high compared to traveler. The problem is the denominator of the pregnant mother for resident. The authors took denominator as the number of pregnant women enrolled in this study. Is it possible to conclude like this?

As mentioned, the proportion of ZIKV infection is higher among resident mothers compared to travellers. This is the main result of the present study, and as mentioned by reviewer 1, “The finding that the risk for pregnant residents is higher than for travellers may seem obvious due to longer exposure to infected mosquitoes, but it is certainly valuable to be confirmed”.

Nevertheless, we agree that this represents the risk among pregnant women seeking medical care, which may underestimate the denominator and consequently overestimate the risk. Nevertheless, most of our patients from endemic countries came from French Guinea, where systematic screening was implemented and therefore, we believe that our denominator very likely offers a good representation of the general population of pregnant women.

Round 2

Reviewer 3 Report

I accept revised manuscript.